# Measuring the conservation attitudes of local communities towards the African elephant *Loxodonta africana,* a flagship species in the Mara ecosystem

**Nyumba Tobias Ochieng**[1,2,3]*, **Kimongo Nankini Elizabeth**[4], **Leader-Williams Nigel**[3]

**1** African Conservation Centre, Nairobi, Kenya, **2** Institute for Climate Change and Adaptation, University of Nairobi, Nairobi, Kenya, **3** University of Cambridge, Cambridge, United Kingdom, **4** Kenya Agricultural and Livestock Research Organisation, Ruiru, Kenya

* tnyumba@uonbi.ac.ke

## Abstract

Gaining insights into local people's views, values and preferences for different conservation management options are increasingly gaining importance among conservationists and decision-makers. This can be achieved through the assessment and understanding of conservation attitudes and perceptions of rural communities including demographic characteristics predicting the attitudes to design and implement conservation policies in a more socially acceptable manner. In this study, we developed and validated user-friendly indices to measure attitudes towards the African elephant, a flagship species and its conservation in Trans Mara District. An iterative item reliability analysis was executed on household data from a random sample of 367 respondents using Cronbach's Alpha in SPSS. Results yielded two indices; (i) Elephant Attitude Index (EAI); and (ii) Maasai Mara National Reserve Attitude Index (MAI). The EAI had a Cronbach's Alpha coefficient of 0.73 while the MAI had a Cronbach's Alpha coefficient of 0.77. Data analysis revealed that (i) location of residence; (ii) age of respondent; (iii) number of income sources; (iv) gender of the respondent; and (v) benefit reception were the main determinants of EAI and MAI in TM. Our attitude indices can assist conservation practitioners and decision-makers to prioritise resources, on the assumption that high-scoring individuals are more likely to participate in conservation initiatives. We encourage making available different sources of income for residents and working towards improving the involvement of younger people and women in conservation activities in TM.

## Introduction

The assessment and understanding of conservation attitudes and perceptions of local communities are increasingly becoming an integral component of conservation research and management [1,2]. This is because, such assessments provide insights on people's preferences for different management options, support for desired wildlife population sizes, the extent of

**Data Availability Statement:** All relevant data are within the manuscript and its Supporting information files.

**Funding:** NT received funding from the Cambridge Commonwealth Trust through the Churchill/Sidney Sussex Southern African Cambridge Scholarship. Additional funding came from the WildiZe Foundation, the Wildlife Conservation Society (Tellus Leadership Scholarship), the Wildlife Conservation Network (Schink Scholarship for Wildlife Conservation) and Churchill College (Pennett Fund Grant and Lundgren Research Award). The funders had no role in study design, data collection and analysis, decision to publish, or preparation of the manuscript.

**Competing interests:** The authors have declared that no competing interests exist, financial or otherwise.

conflict people are willing to tolerate and the desirability of various wildlife species on private or communal land [3–5].

Conservation is a concept that has been used loosely to imply the responsible stewardship of natural resources to sustain the complex social-ecological systems. Evidence suggests that support for conservation is often compromised when people's wellbeing is threatened particularly through costs associated with conservation such as human-wildlife conflict (HWC) [6]. This is even severe when the species involved is a flagship species such as the African elephant (*Loxodonta africana*) and African lion (*Panthera leo*) [7,8]. Elephants are most commonly linked to some of the most intractable forms of conflict with humans, and therefore carry a historical and eminent burden of unfavourable interactions with people [7]. For example, Barnes [9] reported that locals "feared and detested" elephants in Central African forests. Meanwhile, Zimbabwean farmers displayed "ingrained hostility" to elephants who were the "focus of all local animosity toward wildlife" [10], and rural Ugandans "complained bitterly" about elephants, except where they had been eradicated [11]. The literature is rife with reports of damage to crops and property by elephants, resulting in massive crop and financial losses, loss of human lives and injuries [12–14]. Such losses result in negative and antagonistic attitudes towards elephants and conservation initiatives in general and placing a higher political profile on elephants than other wildlife species [e.g. 3,15].

The protected areas (PA) approach to conservation is still expanding particularly in response to species extinction and habitat loss. By 2014, the World Database on Protected Areas report established that PA coverage had increased to 20.6 million sq. Km or 15.4% of the terrestrial earth surface including inland waters. Some 14.7% of these PAs were found in developing countries, particularly in Africa [16]. Although some PAs were designated based on ecological considerations of the larger landscape, evidence suggests that such PAs are not completely isolated ecological entities but overlap with the surrounding inhabited countryside [17–19]. Furthermore, wildlife migration patterns and foraging needs often take animals outside of PAs, onto communal lands, and even across international borders [20]. Consequently, the ability of PAs to address the disconnect between nature conservation and the pursuit of wellbeing has been widely questioned [17,21].

In response to the challenges of PAs, benefit-based approaches to conservation such as community-based conservation (CBC) have been adopted. In this regard, the benefits are considered vital motivational factors for local people to change their attitudes, support conservation efforts, and align their behaviours with conservation goals [3]. Although this notion is premised on the assumption that improving people's wellbeing and providing economic alternatives will improve their attitudes toward conservation, it remains contested among conservation practitioners [3,22]. In addition to conservation costs and benefits, there is burgeoning literature on what influences attitudes and perceptions toward conservation. The most common socio-demographic factors such as wealth, ethnicity, gender, education, the size of household, occupation and age are the most frequently cited [1,3,23–25]. Consequently, there is a consensus that the ecological reasons alone are insufficient in understanding people's attitudes and ensuring support for species and habitat conservation.

## Measuring attitudes and perceptions

Attitudes can be defined as an individual's summary assessment of the degree of favourability, or not, toward one or more concepts, object or stimuli, and the possible conduct and behaviour [26,27]. This assessment includes their beliefs about such concept, object or stimuli as well as a positive and negative evaluation of that belief as shaped by their experiences and perceptions [28,29]. Attitudes can be considered as constructs that facilitate the understanding of

peoples' decision-making and behaviour, hence a theoretical creation based on observations [30]. According to the attitude theory, past negative experiences with an object typically foster negative attitudes towards that object [26]. Thus, negative interactions with conservation activities and wildlife species are expected to contribute to negative attitudes towards the activity and the species in question.

Conservation attitudes are complex and multi-faceted and are influenced by a broad range of social, cultural, political and economic factors as well as any personal experiences of conservation initiatives [3,23,25]. Consequently, there is a need to approach conservation attitude assessment through a framework that recognises the complexity of social-ecological systems by integrating ecological, economic and social perspectives using concepts and methods from a range of disciplines [6,23,31]. The assessment can be operationalised in empirical research through a measurable or observable multi-item index or single-item scales. An index is a measure that combines several distinct indicators of a construct into a single score while a scale is a measure, which captures the intensity, direction, level or potency of a variable construct, and arranges responses or observations on a continuum [30]. Although frequently used, single-item scales usually report low reliability scores and might be difficult to assess. However, an index captures the multidimensionality of the attitudinal construct and can improve the ability to predict behaviour including the inconsistency between attitudes and behaviour [32]. There is a wide range of attitudinal studies in the literature, however, most of the studies have not used objective, quantitative measure, such as an index, to evaluate conservation attitudes of local communities. Conservation attitude researchers have used proxies such as protected area, wildlife species, or a management institution to understand people's attitudes towards the concept [33]. In this study, we measured conservation attitude by using the "elephant" as the proxy for wildlife species and "Maasai Mara National Reserve" as the proxy for a conservation approach. The principal aim of this study was to identify and quantify the conservation attitude of residents through a multi-item index and to identify the underlying components of conservation attitude, including demographic variables.

## Materials and methods

This study was approved by the Ethics Review Group of the University of Cambridge, and the protocols used in the study were approved by the National Council for Science and Technology and Innovation of the republic of Kenya (Permit No. NACOSTI/P/14/0362/2798) and the Kenya wildlife Service (Permit No. KWS/BRM/5001). A total of 367 local residents were interviewed between 2014 and 2015 in the Mara Ecosystem, Kenya and informed consent was sought according to the University of Cambridge Research Ethics guidelines and strategies aimed at minimizing harm to the subject.

### The study area and data collection

Between January and May 2015, we conducted an attitude survey among local community members living in Trans Mara district (TM), adjacent to the world-famous Masai Mara National Reserve (MMNR) in the south-west of Kenya bordering Tanzania at $0°50'–1°50'S$, and $34°35'–35°14'E$ (Fig 1). TM covers approximately 2900 km$^2$ with more than half (2200 km$^2$) being unprotected and forested thereby providing refuge, dry season grazing and dispersal area for a resident, unprotected population of 200–300 elephants [34,35]. Approximately 520 km$^2$ of TM is covered by the Mara Conservancy accounting for 20% of the area of the district, and about 32% of the MMNR [14,36]. The district has five administrative divisions of Lolgorian, Kilgoris, Pirrar, Keyian and Kirindon, with 32 locations and 58 sub-locations. The dominant topographical feature is the Siria Escarpment, formed as a result of faults in the old

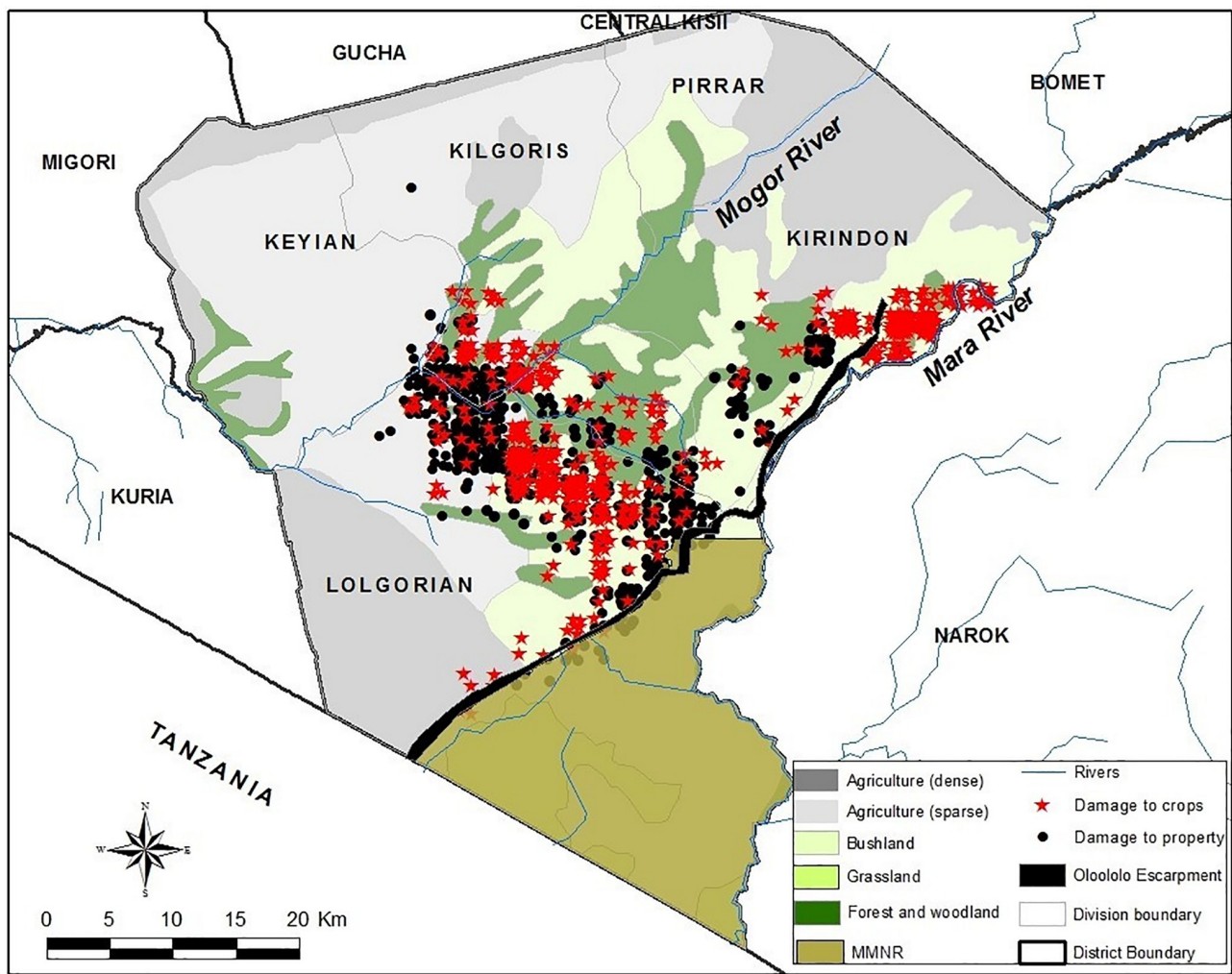

**Fig 1. Trans Mara District showing land use and land cover, administrative divisions and spatial distribution of HEC (source Nyumba et al., 2020 [53]).**

metamorphic rocks [37]. The escarpment, also known to the Maasai as Oloololo (*corner of the earth*) [38], falls away sharply down to Mara River and the MMNR where it plays a significant role in the distribution of wildlife in MMNR through natural corridors along and up its length [36]. Although the divisions fall within the same landscape, they differ in terms of the human population, social-economic activities and prevalence of human-elephant conflicts. Whereas Pirrar and Lolgorian divisions have reported higher human population expansions over the last ten years (2009–2019), Kirindon, Kilgoris and Keyian have reported smaller growth [39]. Meanwhile, crop farming and livestock keeping dominate the highlands of Kilgoris, Pirrar and Kirindon, characterised by rich volcanic and black loam soils and good rainfall that favours the growth of both cereal and cash crops [40]. Wildlife-based tourism in the Mara ecosystem accounts for over 18% of the annual tourist visits to Kenya and is worth an estimated US$15–20 million [41]. Lolgorian and Kirindon divisions account for a range of private and communal wildlife-based enterprises including campsites, tented camps, airstrips, balloon safaris, and lodges [34]. Meanwhile, several sand harvesting points, quarries and small-scale gold mining are found in Kilgoris and Lolgorian divisions [42]. Forest cover in TM has undergone massive

decline due to overexploitation and conversion for agriculture and settlement. The forests in TM play an important socio-cultural role amongst the Maasai especially the provision of shade for council meetings of Maasai elders, provision of fuel, medicine, timber and shafts for spears and arrows, walking sticks, charcoal and fencing posts. In addition, they have provided a key dry season dispersal area and grazing ground for both livestock and wildlife [34]. Land use patterns in TM have changed remarkably quickly, and natural habitats available for elephant conservation have undergone a marked shift [43,44]. This has resulted in a distinctly spatially distributed HEC in the landscape with Kirindon and Lolgorian divisions being the most affected (Fig 1).

A sample of 376 households was selected based on a combination of multi-stage and simple random sampling techniques [45] for our household interviews. We used an inventory of households from the target study sites based on the 2009 population census records [46] and updated by research assistants with support from the local district administration representatives at the sub-location level. This resulted in 17,217 households that formed the sampling frame for this study. We then clustered the households according to sub-location boundaries and allocated each sub-location a proportion of the sample target based on the proportional random sampling (PRS) consideration [47]. We drew a random sample of 376 households from within each cluster through a simple random selection technique using a random number table [48]. This, we considered adequate within a margin of error of 5% and a 95% confidence level (Table 1). We administered the questionnaires to household heads or other adult members (>18years) in the absence of the household head. In all, 367 questionnaires (response rate: 97.60%) were fully executed. The household survey recorded household socio-demographic data, assessment of one's quality of life and attitudinal responses to 16 Likert-type items (S1 Appendix). For each item, respondents were asked to indicate their level of agreement on a 5-point scale, with, 1 = strongly agree; 5 = strongly disagree.

## Attitude index construction methodology

To measure the attitudes toward elephants and elephant conservation in TM, we developed indices for attitude toward elephants (Elephant Attitude Index, EAI) and attitudes toward MMNR (MMNR Attitude Index, MAI) following Babbie [30] based on the 16 Likert-type items. The attitude indices were tested for content validity, face validity, and construct validity [30]. Content validity refers to how much a measure covers the range of meanings included within the concept. This was established by clearly defining the construct of conservation attitude at the outset of the study through literature review. Face validity refers to the extent to which empirical measures may or may not conform to our common understandings and individual mental images concerning a particular concept. We worked with local experts, researchers and practitioners to scrutinise and review the dimensions. Construct validity refers to the logical relationships among variables, which can be statistically investigated. We established

**Table 1. Distribution of samples from each sub-location.**

| Division | No. of Sub-locations | No. of Households | No. of Respondents |
|---|---|---|---|
| Lolgorian | 11 | 6048 | 152 |
| Kirindon | 7 | 4299 | 98 |
| Kilgoris | 4 | 2439 | 58 |
| Keyian | 2 | 2205 | 56 |
| Pirrar | 2 | 477 | 12 |
| **Total** | **26** | **17217** | **376** |

this using Cronbach's alpha and item analysis, yielding a single index with a moderate level of internal reliability. We computed the two indices as follows in SPSS following Cahyat *et al.* [49]:

$$Attitude\ Index = \frac{(Average\ of\ item\ set\ scores) - Minimun\ score}{Maximum\ score - Minimum\ score}\ x100$$

The minimum and maximum scores were derived from the lowest possible score (1) and the highest possible score (5) in the 5-point Likert item scores respectively. The attitude indices scores ranged from 0 to 100 with 0 representing the most negative attitude and 100 representing the most positive attitude. We created score bins for each index to represent negative, neutral and positive attitude.

### Other statistical analyses

The final data were numerically coded and transferred to SPSS 23.0 [50] for statistical analysis. We compared mean conservation attitude scores for EAI and MAI between the Maasai and non-Maasai and among different divisions in TM. We developed regression models for EAI and MAI and multiple potential socio-demographic variables: two were continuous integer variables (age and household sizes); five were categorical variables, three of which were binary (gender of the respondent (male = 1/female = 2)); benefit reception (yes = 1/no = 0), human-elephant conflict (yes = 1/no = 0); and two multinomial variables (number of income sources and living in any one of the five divisions in TM). For the division variable, we created dummy variables for Lolgorian, Kilgoris, Kirindon, Keyian and Pirrar with "0" for not living in the division. However, Pirrar Division was omitted from the model because it had perfect collinearity with Lolgorian, Kilgoris, Kirindon and Keyian divisions. Although we initially included Lolgorian division in the model, it had a higher variance of inflation (10.7) which indicated it could lead to multicollinearity problems and was equally omitted from the final model. We used chi-square, Tukey post-hoc tests, independent-samples t-test (using Levene's test for equality of variances), univariate analysis of variance (ANOVA), Spearman's correlations and general linear regression modelling. All tests were two-tailed, and significance was defined as $p < 0.05$, while $p$ values of $< 0.1$ were considered to indicate trends that may be worthy of future investigation.

## Results

### Attitude index validity

Seven items directly related to the MMNR while four were linked to elephants. Five other items were evaluative assessments of the concepts addressed by the other items and were dropped from the indices. Four of these items provided valuable information about people's perceptions of elephant numbers, their contribution to tourism development and the MMNR (Table 4). One of the items was dropped because its deletion strengthened the Cronbach's alpha of the final elephant attitude index (EAI). The EAI had four items and reported a Cronbach's alpha coefficient of 0.73, whereas the MAI included seven items and had a Cronbach's alpha coefficient of 0.77, suggesting a good internal consistency as measured by George and Mallery [51] guidelines. In addition, all the items had inter-item correlation scores of at least 0.3 [52]. Table 2 summarises the results of the reliability analysis.

### Demographic and socio-economic profile of respondents

A total of 367 respondents comprising more Maasai than non-Maasai ethnic groups were interviewed. Most of the respondents were males. The education levels of respondents and the

**Table 2. Results of the reliability analysis.**

| Attitude toward elephants | Mean | Std. Deviation | Inter-Item Correlation | Cronbach's Alpha (α) |
|---|---|---|---|---|
| Elephants support tourism that brings jobs to the residents | 3.30 | 1.05 | 0.827 | 0.73 |
| Tourist lodges have created business opportunities for the local community | 2.52 | 1.16 | 0.840 | |
| Elephants are responsible for more damage than they are worth | 3.41 | 1.06 | 0.797 | |
| Elephants have become a problem in the community | 2.35 | 1.13 | 0.829 | |
| **Attitude toward MMNR** | | | | |
| MMNR and conservancies have brought positive changes to the community | 2.99 | 1.19 | 0.763 | 0.77 |
| MMNR and conservancies have contributed to education in the village | 2.67 | 1.14 | 0.305 | |
| MMNR and conservancies have caused conflicts among local villagers | 3.09 | 1.16 | 0.740 | |
| I do not support the work of the conservancies | 3.06 | 1.17 | 0.354 | |
| I would be happier if the conservancies were not there | 3.04 | 1.17 | 0.393 | |
| MMNR and conservancies do not protect elephants | 3.41 | 1.07 | 0.456 | |
| MMNR and conservancies do not benefit anyone in the village | 3.11 | 1.12 | 0.777 | |

proportion of household sizes varied between Maasai and non-Maasai ($\chi^2 = 40.5$, $df = 3$, $p<0.001$ and $\chi^2 = 319.3$, $df = 1$, $p<0.001$, respectively) and among the divisions ($\chi^2 = 172.7$, $df = 1$, $p<0.001$) in TM (Table 2). Keyian division had the fewest at 2.6 persons per household and Lolgorian division had the most at 5.3 persons per household. Other demographic characteristics are shown in Table 3.

## Socio-economic activities

Most of the respondents engaged in crop farming (86.0%, n = 316), followed by business and employment (13%, n = 35) including manual or unskilled labour, businesses and teaching. However, 4.0% (n = 16) of the respondents stated that they did not engage in any of the

**Table 3. Generality of respondents.**

| Demographic variable | Category | No. of Respondents by Gender | | No. (%) |
|---|---|---|---|---|
| | | **Female** | **Male** | |
| **Age** | 18–25 | 6 | 14 | 20 (5) |
| | 26–35 | 11 | 53 | 64 (17) |
| | 36–45 | 18 | 84 | 102 (28) |
| | 46–55 | 16 | 88 | 104 (28) |
| | 56> | 7 | 70 | 77 (22) |
| **Gender** | Male | | | 309 (84) |
| | Female | | | 58 (16) |
| **Ethnicity** | Maasai | | | 268 (73) |
| | Kalenjin | | | 83 (24) |
| | Abagusii | | | 11 (3) |
| | Luo | | | 2 (1) |
| | Luhyia | | | 2 (1) |
| | Turkana | | | 1 (1) |
| **Education** | | | **Maasai** | **Non-Maasai** |
| | None | | 117 (44) | 14 (14) |
| | Primary | | 77 (26) | 59 (60) |
| | Secondary | | 55 (21) | 19 (19) |
| | Tertiary | | 25 (9) | 7 (7) |

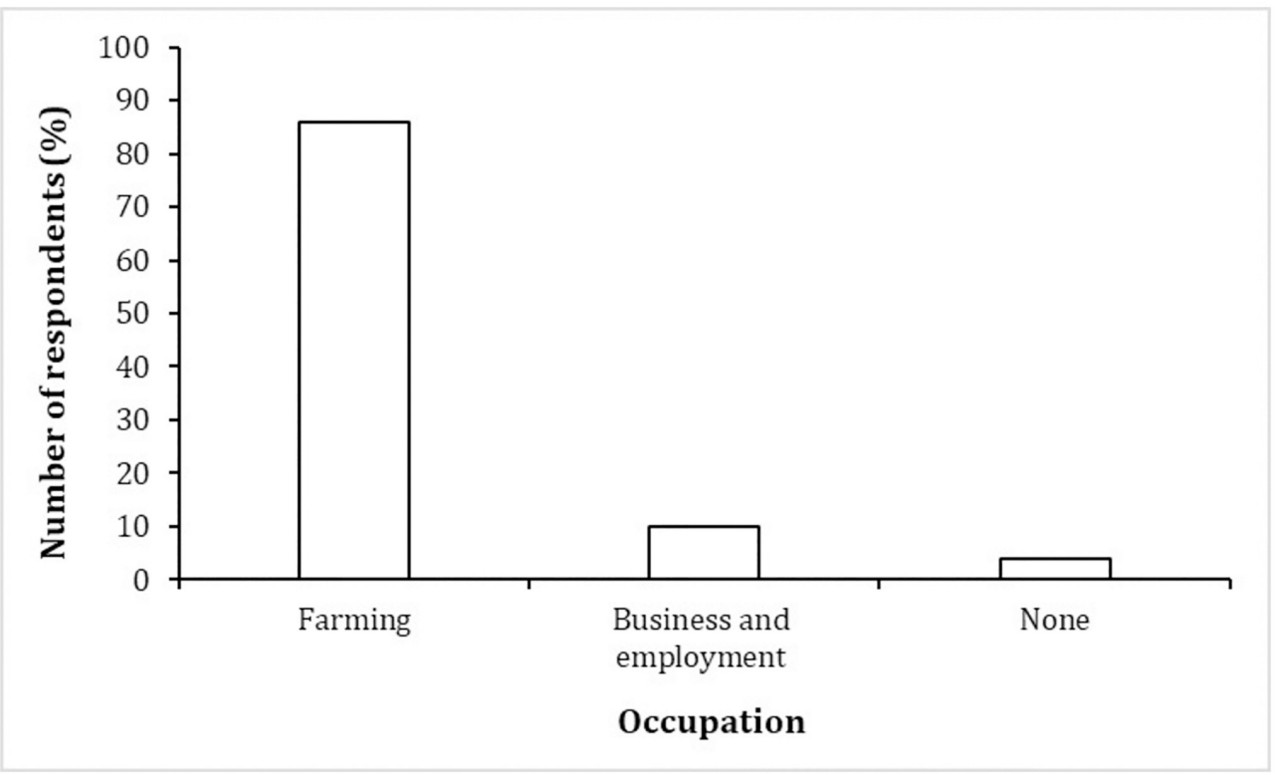

**Fig 2. Socio-economic activities in TM.**

activities (Fig 2) and this was attributed to being in school or college. Most respondents culti-vated crops for subsistence use (24.0%, n = 75) followed by commercial (17.0%, n = 53) and both subsistence and commercial purposes (59.0%, n = 188). Farm sizes ranged between 4ha and 6ha with only 4.1% (n = 13) of the farmers cultivating over 7ha. The rest of the respon-dents who were not engaged in farming cited fear of elephants (59.0%, n = 30), followed by the desire to create space for livestock grazing (29.0%, n = 15) and lack of interest in farming (12.0%, n = 6) as the main reason for not farming. Incidentally, most (67.0%, n = 34) of the respondents not engaged in farming declared their intentions to begin farming within the next five years.

Increasingly, more Maasai (85.0%, n = 229) were engaged in farming. According to local leaders and conservationists in TM, farming has expanded rapidly with increasing uptake of crop farming by the Maasai, perhaps as a diversification strategy into agro-pastoralism. For example, areas like Keyian Division have seen a rapid expansion of sugarcane plantations, especially after the commissioning of the Trans Mara Sugar Company Limited in November 2011, a move that has "*seen traditional Maasai pastoralists increasingly settling down to benefit from the financial opportunities from the sugar company*" (Personal observation and conversa-tion with the Trans Mara sub-county secretary for agriculture).

## Local perceptions of costs and benefits associated with elephants

Our results indicate that half of the respondents felt that elephants were responsible for more damage than they were worth. Furthermore, although more people (46.6%) liked elephants in TM, more than half (52.9%) felt that the elephant population was too high. In terms of benefits, more respondents (42.8%) felt that elephants supported tourism that brought jobs to the

**Table 4. Responses to attitudinal statements.**

| Attitudinal Statements | Agree | Neutral | Disagree |
|---|---|---|---|
| **Attitude toward elephants** | (%) | (%) | (%) |
| Elephants support tourism that brings jobs to the residents | 38.4 | 34.3 | 27.2 |
| Tourist lodges have created business opportunities for the local community | 69.8 | 12.8 | 17.4 |
| Elephants are responsible for more damage than they are worth | 49.9 | 33.8 | 16.3 |
| Elephants have become a problem in the community | 50.7 | 22.6 | 26.7 |
| **Attitude toward MMNR and conservancies** | (%) | (%) | (%) |
| MMNR and conservancies have brought positive changes to the community | 52.9 | 19.6 | 27.5 |
| MMNR and conservancies have contributed to education in the village | 60.5 | 15.0 | 24.5 |
| MMNR and conservancies have caused conflicts among local villagers | 37.6 | 35.1 | 27.2 |
| I do not support the work of the conservancies | 38.7 | 28.3 | 33.0 |
| I would be happier if the conservancies were not there | 37.3 | 24.3 | 38.4 |
| MMNR and conservancies do not protect elephants | 34.3 | 31.3 | 34.3 |
| MMNR and conservancies do not benefit anyone in the village | 26.7 | 36.2 | 37.1 |
| **Not included in any of the indices** | (%) | (%) | (%) |
| Elephants support tourism that brings revenue to the community | 42.8 | 27.0 | 30.2 |
| Elephants are too many | 52.9 | 27.5 | 19.6 |
| Tourist lodges are good for the village | 36.0 | 31.6 | 32.4 |
| I live better because of the conservancies | 47.7 | 31.3 | 21.0 |
| I like elephants | 46.6 | 33.2 | 20.2 |

community. However, at the household level, most respondents (68%) perceived no benefits from elephants or elephant conservation in TM. The results showed that significantly ($\chi^2 = 22.5$, $df = 4$, $p < 0.001$) more respondents from Kilgoris (93.0%) and Pirrar (90.0%) divisions did not receive any benefits compared to Lolgorian (62.0%), Kirindon (67.0%) and Keyian (59.0%) divisions (Table 4).

## Attitude towards elephants and Maasai Mara National Reserve

Ethnic background, Maasai *vs* non-Maasai did not determine attitudes towards elephants in TM ($F_{(1,360)} = 0.6$, $p = 0.425$) and the MMNR ($F_{(1,360)} = 0.1$, $p = 0.916$). This, despite the fact that the Maasai in TM being predominantly pastoralists, have been considered tolerant to elephants than other inhabitants of the district who are predominantly farmers. Further results showed that even when respondents experienced HEC, there was no significant difference ($F_{(1,360)} = 0.7$, $p = 0.420$) in their attitudes toward elephants and Maasai Mara National Reserve ($F_{(1,360)} = 0.5$, $p = 0.475$). However, attitudes towards elephants differed significantly across administrative divisions ($F_{(4,360)} = 3.9$, $p = 0.004$). In particular, residents from Lolgorian (46.1 ±22.3) and Kilgoris (42.9±19.6) divisions held more unfavourable attitudes towards elephants compared to those from Pirrar Division (55.0±10.4). Similarly, respondents from Lolgorian (48.4±17.3) and Kilgoris (47.3±14.8) divisions showed more ($F_{(4,360)} = 9.4$, $p < 0.001$) unfavourable attitudes toward MMNR compared to those from Kirindon (55.4±18.3), Keyian (70.6 ±19.7) and Pirrar (57.7±8.9) divisions. Finally, respondents from Kirindon Division held more unfavourable attitudes toward MMNR (55.4±18.3) compared to those from Keyian Division (70.6±19.7).

## Determinants of attitude towards elephants and MMNR in TM

We ran a linear regression analysis to assess the effects of multiple potential explanatory variables on the elephant attitude index (EAI) and the MMNR attitude index (MAI). The result of

**Table 5. Regression results for elephant attitude index (EAI).**

| Variable | B | SE | β | t | p |
|---|---|---|---|---|---|
| Elephant attitude index (EAI) | 16.14 | 3.73 | 0.00 | 4.33 | < .001 |
| Kilgoris division | 5.17 | 1.40 | 0.18 | 3.68 | < .001 |
| Age of respondent | 0.71 | 0.08 | 0.43 | 8.90 | < .001 |
| Kirindon division | -2.82 | 0.75 | -0.18 | -3.68 | < .001 |

*Note.* $F_{(3,363)}$ = 31.54, $p$ <0.001, $R^2$ = 0.21.

the model with EAI as the dependent variable was significant ($F_{(3,363)}$ = 31.6, $p$<0.001) and produced a goodness of fit of 21% of observed to expected values. The results showed that residing in Kilgoris Division and age of respondent were the main determinants of favourable attitudes towards elephants, whereas residing in Kirindon Division increased unfavourable attitudes toward elephants. This result is not surprising since the Kirindon division borders MMNR and supports some of the elephant refuges within the remnants of the forests in TM. The division has both positive and negative direct interaction with elephants, and residents have raised issues concerning their experiences of HEC [53]. Table 5 summarises the results of the regression model.

On the other hand, using the MMNR attitude index (MAI) as the dependent variable reported a significant ($F_{(7,359)}$ = 432.0, $p$<0.001) relationship with the goodness of fit of 82.0%. Our results showed that an increase in the number of income sources, respondent's age and residing in Pirrar Division significantly led to favourable attitudes toward MMNR. In contrast, residing in Lolgorian and Kirindon divisions decreased MAI scores. Interestingly, receiving benefits from conservation also decreased MAI scores. This is surprising since the community and the MMNR have a benefits-sharing agreement in place. Under the scheme, communities living within 5km of the reserve boundary are entitled to receive 19% of MMNR revenues as compensation. This is accessed and utilised through investment in school infrastructure, bursaries and other social amenities [54]. However, existing literature provides evidence of challenges with the distribution of the benefits leading to a disconnect between local communities and conservancy management, and thus affecting the social relationships others include lack of transparency, accountability and clear structures of participation by stakeholders leading to suspicions and lack of support for what locals considered a "neo-colonialist" plot to expropriate their land [14,34,54]. The negative attitudes in the current study could, therefore, be linked to issues of access and equitable distribution of such benefits. Table 6 summarises the results of the regression model.

**Table 6. Regression results for Maasai Mara National Reserve index (MAI).**

| Variable | B | SE | β | t | p |
|---|---|---|---|---|---|
| MMNR attitude index (MAI) | 6.36 | 3.88 | 0.00 | 3.38 | < .001 |
| Lolgorian | -16.91 | .988 | -.501 | -17.12 | < .001 |
| Age of respondent | 1.19 | 0.03 | 0.90 | 36.97 | < .001 |
| Pirrar | 1.79 | .380 | .088 | 4.73 | < .001 |
| Number of income sources | 2.88 | .533 | .117 | 5.42 | < .001 |
| Benefit reception | -1.39 | .634 | -.039 | -2.20 | .028 |
| Kirindon | -7.29 | .323 | -.580 | -22.58 | < .001 |
| Gender | -2.24 | .275 | -.191 | -8.13 | < .001 |

*Note.* $F_{(7,359)}$ = 432.01, $p$ <0.001, $R^2$ = 0.82.

## Discussion

The results of this study helped in the identification of factors that influence the attitudes of community members toward elephants and the MMNR and conservancies in TM. The use of the MMNR and conservancies as proxies to assess attitudes was based on the consideration that they provide the institutional framework for participation in land-use and wildlife management decisions, especially in shared landscape and community areas bordering protected areas [33]. The results of this study identified age and gender of the respondent, number of income sources, and location of residence, in this case, administrative divisions as significant factors influencing attitudes toward conservation areas and elephants. This partly reflects findings from previous research linking various socio-demographic factors such as education, age, or gender to attitudes toward conservation [55,56].

The Maasai community have historically depicted a unique social, economic and cultural orientation favourable to conservation [36,57]. However, recent accounts point to increasing social-economic and cultural integration with other communities in TM. Our results represent a new outcome where there is no distinction in conservation attitudes based on ethnic orientation. Nevertheless, different divisions reported different scores on conservation attitudes based on their proximity to the elephant conservation area and hence the likelihood of experiencing HEC. Residents in areas where HEC incidents are prevalent are likely to display more unfavourable attitudes toward the elephants and their conservation [58]. When analysed alongside other factors, the results further suggest that HEC did not influence conservation attitudes, rather higher wellbeing, advancement in the age and location of residence far away from the MMNR positively influenced conservation attitudes whereas the location of residence closer to the MMNR and gender reduced the conservation attitudes. This confirms the complex and multifaceted nature of attitudes towards elephants and MMNR. This study identified four factors that contributed to the supportive attitudes toward conservation.

### Factors leading to favourable conservation attitudes

Individuals with diverse sources of income tend to have more favourable conservation attitudes than those with fewer sources of income. This is because, diverse sources of income spread out the risk associated with costs of conservation such as damage to crops and property, loss of livestock, restriction of movement and competition for resources with wildlife. In TM, locals engage in both livestock production and crop farming, employment and business as part of their livelihood and income diversification [34]. Previous research suggests that economic activities affect attitudes and perceptions of local communities towards wildlife conservation [59]. For example, Akama *et al*. [60] established that individuals engaged in non-farm activities in Kenya favoured conservation compared to those who depended solely on crop production for their livelihood options. Similar results were presented by Newmark *et al*. [61] and Infield [55], from other areas of East Africa. The favourable conservation attitudes in TM could be attributed to both the direct and indirect income generation opportunities from the reserve such as employment in the catering, administrative and tour operations function and business opportunities for the sale of food products to the lodges, as well as Maasai cultural items such as embroidery and woodcarvings. When individuals derive direct personal benefit from conservation, they tend to favour and support conservation initiatives. This is because they derive opportunities for direct personal benefit and this is more important than indirect benefits through social investments at the community level [62]. The findings concur with Gillingham and Lee [63] and extend the findings of Infield [55] that households that benefited from the PA held positive attitudes towards conservation than households that did not in Natal, South Africa.

Local communities living along the MMNR boundary are entitled to 19% of tourism revenues as compensation under a benefit-sharing policy [54]. The revenue is mainly invested in educational infrastructure, scholarships and other provisions through a locally elected committee. Furthermore, the reserve has invested in the provision of security, direct employment of early childhood education teachers and compensation for domestic animals killed by predators [64]. This reflects the views of Fiallo and Jacobson [33] that perceived personal benefits must outweigh perceived disadvantages to engender positive attitudes towards conservation. TM has several organisations with different development and conservation initiatives which might make the locals less negative towards elephants and conservancies. Apart from the projects initiated by the MMNR, the WWF Kenya county office in Trans Mara has initiated HEC mitigation projects in HEC hotspots in TM district. Nyumba [35:Ch 4] reports that local communities in TM acknowledged the contribution of HEC mitigation initiatives to awareness about elephant conservation in TM. This is important since it creates a conducive environment to participate in income-generating programmes that might improve their socio-economic conditions and hence overall wellbeing.

In TM, conservation agencies have attempted to address the costs and benefits from conservation through community projects. For example, the WWF Kenya Office in TM has worked with communities to reduce HEC to increase opportunities for locals to participate in economic activities, and initiated reforestation programmes to improve the quality of the environment. Meanwhile, the MMNR has supported the improvement of education, infrastructure, income generation, security and employment to improve access and quality of education, health, transport and communication services. Our results indicated that wellbeing was positively correlated to conservation attitudes. A study in the Caprivi region of Namibia established that locals with higher wellbeing scores were more favourable towards conservancies and tourism activities compared to those with lower wellbeing scores regardless of employment in tourism [65]. In a landscape of competing interests and needs, the desire to meet basic needs is prioritised before higher needs such as support for community-wide initiatives including conservation and development as explained by the needs theories [66,67].

Location of residence relative to the conservation areas determines the level of interactions between people and elephants or conservation activities. The results of this study suggest that the location of residence, in this case, the administrative division had a significant effect on the conservation attitudes. People living in Kilgoris and Pirrar divisions, which are far away from the park boundary showed favourable conservation attitudes. The two divisions are located approximately 30km to the west of the MMNR. Similar results were found in Zimbabwe where people living closer to the PAs displayed negative attitudes toward PAs despite access to natural resources [2]. Furthermore, although residents of Kilgoris and Pirrar reported conflict with elephants, their interaction was limited to soft boundaries along the edge of the forest, with seasonal elephant presence. According to Guerbois *et al.* [2], residents sharing hard boundaries with the PAs tend to be more negative compared to those sharing soft boundaries.

Finally, the contribution of age to conservation attitudes in this study was significant but did not support previous findings such as 2,25 and [67–69] who established a strong negative correlation between age and attitudes. In particular, the studies established that younger members of the community tend to favour conservation and are more tolerant of conflict species. Instead, our results indicate that advancement in age led to a more favourable conservation attitude. Several reasons could explain this. First, the Maasai community is largely patriarchal where the head of the household, usually an older male, dominates the spheres of decision making from the household to the community level. They tend to participate in various community-based decision-making institutions as representatives and hence access to the benefits

and information on the conservation initiatives, including employment opportunities, scholarship disbursements, allocations for school infrastructure and other benefits to the community.

Second, the Maasai have ancestral and totemic respect for certain wildlife species and have developed ways to coexist with dangerous wildlife which is then passed on to the younger generations. However, recent social-cultural changes have led to a breakdown in the traditional knowledge being passed on to the younger generation [70]. Similar changes were found to influence attitudes toward crocodiles in Uganda where the younger members of the community lacked indigenous knowledge on the role of crocodiles in the natural ecosystem and simply viewed crocodiles as a threat and as a source of hardship because they attacked livestock and competed for fish [71]. In addition, most of the younger members of the Maasai community have taken up farming and are clearly inexperienced in managing crop damage by elephants and hence the recent increase in incidents of crop raids in TM [34]. Previous studies have established that immigrants and new members of the community tend to have negative attitudes towards problem animals due to lack of experience with conflict mitigation [36,72,73].

Third, TM has undergone a rapid shift in education with more Maasai youths getting formal education and training. Consequently, many younger members of the community have some skills but are not gainfully employed. This is partly due to the few opportunities available in the MMNR and conservancies which cannot absorb most of the graduates and partly due to the few opportunities available from the government and private sector in the community. Coupled with the hogging of opportunities by the leaders for their relatives or supporters, the younger members have little or no access to opportunities. Furthermore, the existing conservation and development organisations such as WWF, KALRO, and World Vision in TM can only offer few positions such as community scouts, field enumerators and outreach officials on a seasonal basis. However, like the opportunities in the MMNR and lodges, the occupants of these positions are equally determined by the community leaders since all or most of the recruitment is done in consultation with the local leaders. Furthermore, some of the skills of the younger educated members of the community do not match the available opportunities. Consequently, the unfavourable support for conservancies and elephants in TM by the young generation could be attributed to frustrations borne out of the unmet expectations of better employment, social exclusion, access to services and income and wealth after school and the decreasing importance of tradition and cultural transmission of information.

## Factors leading to unfavourable attitudes

The fact that people living closer to the hard boundaries of PAs do have frequent interaction with wildlife and conservation activities is intuitive. The results of this study suggest that people living in Kirindon and Lolgorian divisions, which share boundaries with MMNR and have a large proportion of the remaining elephant refuge in TM tend to have an unfavourable conservation attitude. Various studies have linked the unfavourable attitudes of communities closer to protected areas towards conservation activities to the direct costs of living with problematic animals and inadequate benefits accrued from the conservation activities [e.g. 2,73–75]. This implies that residents of Kirindon and Lolgorian divisions do face not only the direct impacts of HEC but also experience other indirect and complex impacts of HEC. For example, as a result of frequent crop damage, the general threat from elephants, lack of adequate employment opportunities and restricted access to the reserve, these residents might have felt restricted in undertaking their social-cultural and economic activities to improve their wellbeing.

Gender, on the other hand significantly contributed to unfavourable conservation attitudes, particularly among females. From an evolutionary perspective, the Maasai men were more outgoing hunters and more eager to take risks while women stayed at home taking care of the family and children. Consequently, males tend to be more tolerant and knowledgeable about wildlife species and conservation [36]. Studies elsewhere suggest that the unfavourable attitudes of women towards wildlife could be attributed to a greater apprehension about dangerous species [76,77]. TM has experienced rapid social, economic and political integration. Consequently, more women are now in constant contact with the environment and can contribute to the management decisions, but this has also increased the risk of encountering dangerous animals in the landscape [78,79]. Furthermore, women in TM accompanied their children to school and health centres, went to the markets to buy food for their families, and visited nearby forests to collect firewood and streams for water. The risk of encountering elephants and of being arrested in the forest hinder the access to the essential services which tend to align with activities and social roles in the society [80]. Conservation institutions in TM and the MMNR have made available opportunities for community members to participate in decision-making through representation, and to benefit from employment and participation in development projects. However, most women in TM do not have the desired skills to take advantage of these employment opportunities, whereas those who have some skills cannot be involved because the opportunities are male-oriented and tend to be risky for women. According to Gustafson [79], the perception and interpretation of risks are complex and vary between women and men even when the risk appears to be the same. In addition, Maasai cultural orientation tends to relegate women regarding leadership and representation. Consequently, the unequal power relations and access to opportunities from conservation could be affecting the social wellbeing of women leading to low levels of trust in conservation institutions and wildlife species in TM.

## Conclusions and recommendations

This study has demonstrated that the conservation attitudes in TM did not vary between the dominant Maasai and the other non-Maasai ethnic groups living in the district. This suggests a possible loss of the historically unique social, economic and cultural orientations of the Maasai [36,57] which depicted them as conservation-friendly [81,82]. Instead, conservation attitudes varied based on the location of residence where people living close to the protected areas had unfavourable conservation attitudes compared to those living far away. Finally, future research could adapt and apply the indices developed for measuring the conservation attitudes of residents in TM in other areas. However, there must be a clear understanding of the specific contextual factors of the target area. It is our view that the indices can be used in repeat surveys to determine whether conservation attitudes in TM have changed.

The conservation and management of elephants in TM, and indeed Kenya, has placed elephants at the centre of the "flagship-battleship" debate. While the conservation agencies hold elephants as a flagship for conservation, elephants represent a daily competition over resources and livelihoods with the residents. Addressing these challenges requires innovative and proactive approaches including the preservation and documentation of traditional knowledge and promotion of education and awareness among the younger generations on the value the Masai Mara ecosystem. Identify and promote the implementation of diverse livelihood opportunities including those connecting MMNR with the locals and the efficient and adequate compensation for damages. Conservation actors need to promote participatory and community-oriented HEC mitigation to foster tolerance and co-existence between people and wildlife.

## Supporting information

**S1 Appendix. Household questionnaire.**
(DOCX)

## Acknowledgments

Our thanks to all residents of TM who participated in the household surveys. Our thanks to Professor Sarah White for advice on the development and use of indices. Sarah White and Professor Bill Sutherland and two anonymous referees provided helpful advice on earlier drafts of the manuscript.

## Author Contributions

**Conceptualization:** Nyumba Tobias Ochieng, Leader-Williams Nigel.

**Data curation:** Nyumba Tobias Ochieng, Kimongo Nankini Elizabeth.

**Formal analysis:** Nyumba Tobias Ochieng.

**Funding acquisition:** Nyumba Tobias Ochieng, Leader-Williams Nigel.

**Investigation:** Nyumba Tobias Ochieng, Kimongo Nankini Elizabeth, Leader-Williams Nigel.

**Methodology:** Nyumba Tobias Ochieng, Kimongo Nankini Elizabeth, Leader-Williams Nigel.

**Project administration:** Nyumba Tobias Ochieng, Kimongo Nankini Elizabeth, Leader-Williams Nigel.

**Resources:** Nyumba Tobias Ochieng, Leader-Williams Nigel.

**Software:** Nyumba Tobias Ochieng.

**Supervision:** Nyumba Tobias Ochieng, Leader-Williams Nigel.

**Validation:** Nyumba Tobias Ochieng, Leader-Williams Nigel.

**Visualization:** Nyumba Tobias Ochieng.

**Writing – original draft:** Nyumba Tobias Ochieng.

**Writing – review & editing:** Nyumba Tobias Ochieng, Kimongo Nankini Elizabeth, Leader-Williams Nigel.

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
