## [Decision Letter · Decision Letter 0]

8 Apr 2021

PONE-D-21-07081

Measuring the Conservation Attitudes of Local Communities Towards the African Elephant Loxodonta africana, A Flagship Species in the Mara Ecosystem

PLOS ONE

Dear Dr. Nyumba,

Thank you for submitting your manuscript to PLOS ONE. After careful consideration, we feel that it has merit but does not fully meet PLOS ONE’s publication criteria as it currently stands. Therefore, we invite you to submit a revised version of the manuscript that addresses the points raised during the review process.

We look forward to receiving your revised manuscript.

Kind regards,

Tunira Bhadauria, Ph.D.

Academic Editor

PLOS ONE

Journal Requirements:

Please provide additional details regarding participant consent. In the ethics statement in the Methods and online submission information, please ensure that you have specified (1) whether consent was informed and (2) what type you obtained (for instance, written or verbal, and if verbal, how it was documented and witnessed). If your study included minors, state whether you obtained consent from parents or guardians. If the need for consent was waived by the ethics committee, please include this information.

Please include your tables as part of your main manuscript and remove the individual files. Please note that supplementary tables (should remain/ be uploaded) as separate "supporting information" files

We note that Figure 1 in your submission contain map images which may be copyrighted. All PLOS content is published under the Creative Commons Attribution License (CC BY 4.0), which means that the manuscript, images, and Supporting Information files will be freely available online, and any third party is permitted to access, download, copy, distribute, and use these materials in any way, even commercially, with proper attribution. For these reasons, we cannot publish previously copyrighted maps or satellite images created using proprietary data, such as Google software (Google Maps, Street View, and Earth). For more information, see our copyright guidelines: http://journals.plos.org/plosone/s/licenses-and-copyright.

4a, You may seek permission from the original copyright holder of Figure 1 to publish the content specifically under the CC BY 4.0 license. 

4b, If you are unable to obtain permission from the original copyright holder to publish these figures under the CC BY 4.0 license or if the copyright holder’s requirements are incompatible with the CC BY 4.0 license, please either i) remove the figure or ii) supply a replacement figure that complies with the CC BY 4.0 license. Please check copyright information on all replacement figures and update the figure caption with source information. If applicable, please specify in the figure caption text when a figure is similar but not identical to the original image and is therefore for illustrative purposes only.

Reviewers' comments:

Reviewer's Responses to Questions

**Comments to the Author**

1. Is the manuscript technically sound, and do the data support the conclusions?

Reviewer #1: Partly

Reviewer #2: Yes

2. Has the statistical analysis been performed appropriately and rigorously? 

Reviewer #1: Yes

Reviewer #2: N/A

3. Have the authors made all data underlying the findings in their manuscript fully available?

Reviewer #1: No

Reviewer #2: Yes

4. Is the manuscript presented in an intelligible fashion and written in standard English?

Reviewer #1: Yes

Reviewer #2: Yes

5. Review Comments to the Author

Reviewer #1: The subject of the research is important in the current context because it throws light on improving attitude of communities for conservation of species and habitats. However, a plentiful of studies of similar nature have been conducted globally and particularly in African protected areas. It is well known that conflict between a local people and a species or a protected area can weaken conservation efforts. Thus, the topic is not novel, yet important for improving conservation prospect of elephants in Trans Mara (TM) district which is adjacent to Masai Mara National Reserve.

Following suggestions are put forward to improve the manuscript;

i. Describe salient features of the five administrative divisions with special reference to natural resources availability, demography etc. for better understanding of the situation and study area.

ii. For questionnaire survey 376 households were selected for household interviews. Authors should explain adequacy of the sample size for a desired level of precision. Further, the authors should mention the number of households surveyed in different divisions of TM and justify the number of households surveyed in the divisions.

iii. Authors should explain the criteria of categorization of Attitude Indices Scores into ‘negative’, ‘neutral’ and ‘positive’ attitude for clarity to readers (Page No. 4, Line nos. 167-169).

iv. Authors are suggested to consider– ‘distance of household / community settlement from the boundary of protected area’ as a variable in the model and see if it improves the model.

v. Have the authors worked out ‘community-forest/park’ interaction in terms of communities’ dependence on the forest areas for Provisioning Services? Direct benefits from natural habitats could be an important determinant of people’s attitudes.

vi. The authors have collected in-depth information on attributes representing socio-economic status of studied households. It is suggested to present key socio-economic attributes in a table and explain them aptly to improve readability and better understanding of these drivers of conservation attitude.

vii. ‘Human wellbeing’ has not been explained in the manuscript but in the Results section association of human wellbeing and conservation attitudes have been presented (page no. 6, line nos. 240-246). The authors have mentioned ‘higher wellbeing scores’ (line no. 246), but how these scores have been arrived at are not described in the manuscript. Secondly, in the same section 0.542** and 0.449**have been shown as a parameter value, but it is not clear what these values are?

viii. Have the authors studies issues related to access and equitable distribution of benefits? (page no.6, line no. 264-265). In absence of any supporting data, it seems to be speculative.

ix. Human-wildlife interaction substantially influences attitude of people; thus it would be good if the authors present the HEC scenario in different divisions of TM.

x. Page No. 9, para. 3: It is clear that opportunities are availed mainly by a select group of people in the society, and the unfavorable conservation attitude of younger generation is due to lesser opportunities and is a reflection of frustration of the youth. Do these factors also get reflected in unfavorable attitude towards governance and society at large, or these are limited only to conservation? The authors should throw light on this aspect.

Reviewer #2: Comments/suggestions to the authors:

1. The abstract has to be more precise.

2. When was the study conducted? Mention in “materials and methods” section.

3. Line no. 179: The number of dummy variables to be used is one less than the total number of categories. In order to simplify, the divisions can be categorized into two zones, a)zone close to MMNR (Lolgorian, Kirindon) and b)zone far from MMNR (Kilgoris, Keyian and Pirrar). Thus two dummy variables can be used for 5 divisions (suggestions are based solely on the map provided). Hence the authors are advised to rerun the regression.

4. The attitude towards elephants (Table no. 1) is broadly tourism based. What about the positive role of the species in the forest ecosystem? Do the locals value the existence of the mammal? These questions were not asked. Conservation of the species should not be solely based on tourism returns. Therefore, a change in the title is suggested. The title can be modified into “Exploring the Socio-Economic Aspect of Conservation Attitude in Different Ethnic Groups towards the African Elephant in the Mara Ecosystem”.

5. The authors can include suggestions like preservation and documentation of traditional knowledge related to wildlife, creation of more jobs connecting MMNR with the locals, proactive government in providing adequate and immediate compensation for damages, alternative livelihoods involving non-conversion of forests, involvement of locals in rescue operations of injured elephants or orphan calves as well as building task force for protection of wildlife. The young generation must be educated how to value the Masai Mara ecosystem. In case of existing agricultural lands, the farmers can be taught to safe-guard their crops using latest techniques (location-specific solutions such as bee-fencing). Adaptation and tolerance are the keys to co-existence.

6. Line nos. 467-468: The term “battle” can be more sensitively replaced by the ecological term “competition”. The last sentence is to be omitted for not generating a negative/destructive attitude towards the species.

6. PLOS authors have the option to publish the peer review history of their article (what does this mean?). If published, this will include your full peer review and any attached files.

Reviewer #1: No

Reviewer #2: **Yes: **Bindia Gupta

---

## [Author Response · Author response to Decision Letter 0]

25 May 2021

All the responses have been made in the rebuttal letter and within the response to reviewers' comment document uploaded.

---

## [Editor Report · Decision Letter 1]

1 Jun 2021

Measuring the Conservation Attitudes of Local Communities Towards the African Elephant Loxodonta africana, A Flagship Species in the Mara Ecosystem

PONE-D-21-07081R1

Dear Dr. Nyumba

We’re pleased to inform you that your manuscript has been judged scientifically suitable for publication and will be formally accepted for publication once it meets all outstanding technical requirements.

Kind regards,

Tunira Bhadauria, Ph.D.

Academic Editor

PLOS ONE

Additional Editor Comments (optional):

I would like to congratulate the authors for having revised manuscript well by incorporated all the comments/ suggestions put forward by the two reviewers. The revised manuscript holds sufficient merit and is scientifically sound enough to be accepted for publication in the Journal. therefore I recommend that manuscript be accepted for publication in the esteemed Journal.
---

## [Editor Report · Acceptance letter]

14 Jun 2021

PONE-D-21-07081R1 

Measuring the Conservation Attitudes of Local Communities Towards the African Elephant *Loxodonta africana*, A Flagship Species in the Mara Ecosystem 

Dear Dr. Tobias Ochieng:

I'm pleased to inform you that your manuscript has been deemed suitable for publication in PLOS ONE. Congratulations! Your manuscript is now with our production department. 

Kind regards, 

on behalf of

Dr. Tunira Bhadauria 

Academic Editor

PLOS ONE